# Explicit and exact travelling wave solutions for Hirota equation and computerized mechanization

**Bacui Li[1], Fuzhang Wang[2]\*, Sohail Nadeem[3]**

**1** Scientific Research Department, Party School of CPC Fushun, Fushun, Liaoning, China, **2** School of Mathematics and Statistics, Xuzhou University of Technology, Xuzhou, China, **3** Department of Mathematics, Quaid-i-Azam University, Islamabad, Pakistan

\* wangfuzhang1984@163.com

**Data Availability Statement:** All relevant data are within the manuscript.

**Funding:** The work was supported by the University Natural Science Research Project of Anhui Province (Project No. 2023AH050314) and

## Abstract

By using the power-exponential function method and the extended hyperbolic auxiliary equation method, we present the explicit solutions of the subsidiary elliptic-like equation. With the aid of the subsidiary elliptic-like equation and a simple transformation, we obtain the exact solutions of Hirota equation which include bright solitary wave, dark solitary wave, bell profile solitary wave solutions and Jacobian elliptic function solutions. Some of these solutions are found for the first time, which may be useful for depicting nonlinear physical phenomena. This approach can also be applied to solve the other nonlinear partial differential equations.

## 1 Introduction

Recently, optical solitons have become the focus of theoretical and experimental research. Optical solitons are temporally localized pulses or spatially bounded homing beams-produced by nonlinear changes in the refractive index of the material caused by the light intensity distribution [1]. In this case, the main model that controls the evolution of the pulse is the famous nonlinear Schrödinger (NLS) equation

$$iq_t + \frac{1}{2}q_{xx} - \sigma|q|^2 q = 0, \tag{1}$$

where $\sigma = -1$ corresponds to the focusing NLS equation, which produces a bright soliton, and $\sigma = 1$ corresponds to the defocusing NLS equation, which produces a dark soliton.

The scalar NLS equation discussed above is a relatively general model which is used to explain various effects in the transmission of optical pulses. It is derived from general assumptions based on the dispersion or diffraction and nonlinear properties of physical systems. However, the NLS model may be insufficient to accurately describe the problem in many cases. For example, the NLS equation can describe the transmission of picosecond optical pulses in optical fibers, but for the transmission of sub-picosecond or femtosecond pulses in optical fibers at high speeds, narrow pulses will produce high-order dispersion, self-

the Horizontal Scientific Research Funds in Huaibei Normal University (No. 2024340603000006).

**Competing interests:** The authors have declared that no competing interests exist.

attenuation effects and intra-pulse Raman scattering. Considering the above-mentioned effect of high-order dispersion and self-attenuation, ultrashort pulses in optical fibers will be governed by the following higher-order nonlinear equations

$$iq_t + \frac{1}{2}q_{xx} + \sigma|q|^2q + i\alpha[\beta_1 q_{xxx} + \sigma\beta_2|q|^2q_x + \beta_3 q(|q|^2)_x] = 0, \tag{2}$$

where $q(x, t)$ is a complex function, $\beta_1$, $\beta_2$ and $\beta_3$ are real parameters related to the third-order dispersion, the self-attenuation effect and the pulse Raman scattering, respectively. In particular, Eq (2) can be reduced to the NLS equation for $\alpha = 0$. In general, Eq (2) is not completely integrable unless certain restrictions are imposed on $\beta_1$, $\beta_2$ and $\beta_3$.

When considering the third-order dispersion and the third-order dispersion attenuation effect, the transmission of ultrashort pulses in the fiber can be controlled by the Hirota equation

$$iq_t + \frac{1}{2}q_{xx} - \sigma|q|^2q + i\alpha(-q_{xxx} + 6\sigma|q|^2q_x) = 0, \tag{3}$$

where $q = q(x, t)$, $(x, t) \in R \times R$, $q_{xx}$ represents group velocity dispersion, $|q|^2q$ represents self-phase modulation, $q_{xxx}$ represents third-order dispersion, $|q|^2q_x$ represents self-attenuation effects, $\alpha$ is a real parameter and $\sigma = \pm 1$.

The nonlinear partial differential equations(PDEs) are different from the linear case, there is usually great difficulty in obtaining the exact solutions. But for the solutions of integrable equations, there are several effective and significant methods which include Darboux transform [2], Bäcklund transform [3], inverse scattering transform method [4], Hirota method [5], Lie group method [6], and auxiliary equation method [7]. Among them, the inverse scattering method can be used to solve the initial value problem and its exact solution can be written for the special case of no reflection coefficient. For the general case, the asymptotic properties or long-time behavior of the solution can be obtained by the nonlinear speed drop method. The Hirota bilinear method is a straightforward method in finding soliton-type solutions based on the Hirota bilinear derivatives. The Darboux transform is a method based on the normative invariance of the integrable equation, which corresponds to the non-potential case of the inverse scattering method. The key idea of the traditional auxiliary equation method is to use the solutions of new first-order nonlinear ordinary differential equation instead of tanh(.) in tanh-function method and extended tanh-function method. By using the extended hyperbolic auxiliary equation method, a series of new travelling wave solutions can be obtained including not only Jacobian elliptic function solutions but also solitary wave solutions and trigonometric function solutions [8].

In this paper, we will mainly consider the defocusing Hirota Eq (3) in the case of $\sigma = 1$

$$iq_t + \frac{1}{2}q_{xx} - |q|^2q + i\alpha(-q_{xxx} + 6|q|^2q_x) = 0, \tag{4}$$

The multi-soliton solutions, breathing sub-solutions and strange wave solutions of the Hirota equation have been studied extensively and deeply. For example, the Riemann-Hilbert method is investigated to construct the bright soliton solution of the coupled Hirota equation [9]. The multi-soliton solutions, breathing sub-solutions, strange wave solutions are constructed by using the Darboux transformation [10]. Bindu et al. considered the coupled Hirota equation and used Painlev'e analysis to obtain the parameter conditions for the existence of bright and dark solitons. The general Lax pairs are constructed by the bilinear method and the dark soliton solution is obtained [11].

Motivated by the above-analysis, a new algebraic method named the auxiliary elliptic-like equation method is proposed in this article. The main idea of the new algorithm is that we present the explicit solutions of the subsidiary elliptic-like equation by using the power-exponential function method and the extended hyperbolic auxiliary equation method. Based on the subsidiary elliptic-like equation and a simple transformation, several new exact solutions to the Hirota equation are worked out, including bright solitary wave, dark solitary wave, bell profile solitary wave solutions and Jacobian elliptic function solutions. These cannot be found in the previous literatures and may be useful for depicting nonlinear physical phenomena. This method can also be applied to other nonlinear PDEs in mathematical physics. The rest of this article is arranged as follows. In Section 2, the extended hyperbolic auxiliary equation method is introduced. In Section 3, the exact solutions of the subsidiary elliptic-like equation are obtained, with the aid of the power-exponential function method and the extended hyperbolic auxiliary equation method. By using a simple transformation and the subsidiary elliptic-like equation, the exact solutions of the Hirota equation are derived in Section 4. In Section 5, the conclusions of this paper can be found.

## 2 Introduction of the extended hyperbolic auxiliary equation method

### 2.1 Steps of the extended hyperbolic auxiliary equation method

**Step 1** For a given nonlinear PDE with one physical field $q(x, t)$ in two variable $x, t$,

$$\Lambda(q, q_x, q_t, q_{xx}, q_{xt}, q_{tt}, \cdots) = 0. \tag{5}$$

Some cases describe the propagation of optical pulses in linear/nonlinear optic fibers.

We seek its travelling wave solution, in the form of $q(x, t) = q(\zeta)$, $\zeta = k(x + lt - \lambda)$, where $k, l$ and $\lambda$ are constants to be determined. The nonlinear PDE (5) is reduced to a nonlinear ordinary differential equation(ODE)

$$\Pi(q, q_\zeta, q_{\zeta\zeta}, q_{\zeta\zeta\zeta}, \cdots) = 0. \tag{6}$$

**Step 2** To seek for the travelling wave solutions of (6), we assume that (6) has solutions in the form of

$$q(\zeta) = a_0 + \sum_{i=1}^{n} \cosh^{i-1} w(\zeta)[a_i \sinh w(\zeta) + b_i \cosh w(\zeta)], \tag{7}$$

where $a_i, b_j(i = 0, 1, 2, \cdots, n; j = 1, 2, \cdots, n)$ are constants to be determined, $\sinh w(\zeta), \cosh w(\zeta)$ are determined by the following hyperbolic auxiliary equation

$$\left(\frac{dw(\zeta)}{d\zeta}\right)^2 = f + g \sinh w(\zeta) \cosh w(\zeta) + h \cosh^2 w(\zeta). \tag{8}$$

By balancing the highest degree linear term and nonlinear term in (6), we can determine the degree $n$.

**Step 3** Substituting (7) along with (8) into (6) and setting the coefficients of $\sinh^p w(\zeta)$ $\cosh^r w(\zeta)$ $(r = 0, 1; p = 0, 1, \cdots, n + 2)$ to zero, we will obtain a set of algebraic equations with respect to the parameters $k, l, \lambda, a_i, b_j(i = 0, 1, 2, \cdots, n; j = 1, 2, \cdots, n)$.

**Step 4** Solving the set of algebraic equations with the aid of the symbolic computation software (Maple), we would end up with the explicit expressions for $k, l, \lambda, a_i, b_j(i = 0, 1, 2, \cdots, n; j = 1, 2, \cdots, n)$.

**Step 5** By considering the value of $f$, $g$, $h$, Eq (8) has many kinds of dark solitary wave, bell profile solitary wave, singular soliton and Jacobian elliptic function solutions [12], which are listed in the following subsection.

## 2.2 Various cases

**Case1** For $f = -2(1 + m^2)$, $g = 2(-1 + m^2)$, $h = 2(1 + m^2)$, we acquire a Jacobi elliptic doubly periodic-type solution

$$\sinh w(\zeta) = \frac{-cn^2(\zeta)}{2sn(\zeta)}, \ \cosh w(\zeta) = \frac{2 - cn^2(\zeta)}{2sn(\zeta)}. \tag{9}$$

**Case2** For $f = 2(-1 + m^2)$, $g = -2$, $h = -2(-1 + 2m^2)$, we acquire a Jacobi elliptic doubly periodic-type solution

$$\sinh w(\zeta) = \frac{-sn^2(\zeta)}{2cn(\zeta)}, \ \cosh w(\zeta) = \frac{2 - sn^2(\zeta)}{2cn(\zeta)}. \tag{10}$$

**Case3** For $f = -2(-2 + m^2)$, $g = -2m^2$, $h = 2(-2 + m^2)$, we acquire a Jacobi elliptic doubly periodic-type solution

$$\sinh w(\zeta) = \frac{-m^2[1 + cn^2(\zeta)]}{2dn(\zeta)}, \ \cosh w(\zeta) = \frac{2 - m^2[1 + cn^2(\zeta)]}{2dn(\zeta)}. \tag{11}$$

**Case4** For $f = 0$, $g = -2m^2$, $h = -2(-2 + m^2)$, we acquire a Jacobi elliptic doubly periodic-type solution

$$\sinh w(\zeta) = \frac{-1 + sc^2(\zeta)}{2sc(\zeta)}, \ \cosh w(\zeta) = \frac{1 + sc^2(\zeta)}{2sc(\zeta)}. \tag{12}$$

**Case5** For $f = -2 + 3m^2 - m^4$, $g = 2(-1 - m^2 + m^4)$, $h = 2(1 - m^2 + m^4)$, we acquire a Jacobi elliptic doubly periodic-type solution

$$\sinh w(\zeta) = \frac{-1 + sd^2(\zeta)}{2sd(\zeta)}, \ \cosh w(\zeta) = \frac{1 + sd^2(\zeta)}{2sd(\zeta)}. \tag{13}$$

**Case6** For $f = -2(1 + m^2)$, $g = 2(-1 + m^2)$, $h = 2(1 + m^2)$, we acquire a Jacobi elliptic doubly periodic-type solution

$$\sinh w(\zeta) = \frac{-1 + cd^2(\zeta)}{2cd(\zeta)}, \ \cosh w(\zeta) = \frac{1 + cd^2(\zeta)}{2cd(\zeta)}. \tag{14}$$

**Case7** For $f = m^2$, $g = 0$, $h = 1 - m^2$, we acquire a Jacobi elliptic doubly periodic-type solution

$$\sinh w(\zeta) = \pm\frac{sn(\zeta)}{cn(\zeta)}, \ \cosh w(\zeta) = \frac{1}{cn(\zeta)}. \tag{15}$$

**Case8** For $f = 1$, $g = 0$, $h = -1 + m^2$, we acquire a Jacobi elliptic doubly periodic-type solution

$$\sinh w(\zeta) = \pm\frac{msn(\zeta)}{dn(\zeta)}, \ \cosh w(\zeta) = \frac{1}{dn(\zeta)}. \tag{16}$$

**Case9** For $f = -1, g = 0, h = m^2$, we acquire a Jacobi elliptic doubly periodic-type solution

$$\sinh w(\zeta) = -\frac{1 \pm dn(\zeta)}{2msn(\zeta)} + \frac{msn(\zeta)}{2[1 \pm dn(\zeta)]}, \ \cosh w(\zeta) = \frac{1 \pm dn(\zeta)}{2msn(\zeta)} + \frac{msn(\zeta)}{2[1 \pm dn(\zeta)]}. \quad (17)$$

**Case10** For $f = -m^2, g = 0, h = 1$, we acquire a Jacobi elliptic doubly periodic-type solution

$$\sinh w(\zeta) = \pm \frac{cn(\zeta)}{sn(\zeta)}, \ \cosh w(\zeta) = \frac{1}{sn(\zeta)}. \quad (18)$$

**Case11** For $f = \frac{1}{4}(-5 + 2m^2 - m^4), g = \frac{1}{2}(-1 + m^4), h = \frac{1}{2}(1 + m^4)$, we acquire a Jacobi elliptic doubly periodic-type solution

$$\sinh w(\zeta) = \frac{[1 \pm dn(\zeta)]}{2sn(\zeta)} \left( -1 + \frac{sn^2(\zeta)}{(1 \pm dn(\zeta))^2} \right), \ \cosh w(\zeta) = \frac{sn^2(\zeta) + [1 \pm dn(\zeta)]^2}{2sn(\zeta)[1 \pm dn(\zeta)]}. \quad (19)$$

**Case12** For $f = 2, g = -2, h = -2$, we acquire a bell profile solitary wave solution

$$\sinh w(\zeta) = \frac{1}{2}[-1 + \mathrm{sec}h^2(\zeta)] \cosh(\zeta), \ \cosh w(\zeta) = \frac{1}{2}[1 + \mathrm{sec}h^2(\zeta)] \cosh(\zeta). \quad (20)$$

**Case13** For $f = -4, g = 0, h = 4$, we acquire a dark soliton wave solution

$$\sinh w(\zeta) = \frac{-1 + \tanh^2(\zeta)}{2\tanh(\zeta)}, \ \cosh w(\zeta) = \frac{1 + \tanh^2(\zeta)}{2\tanh(\zeta)}. \quad (21)$$

**Case14** For $f = 0, g = 2, h = 2$, we acquire a singular soliton solution

$$\sinh w(\zeta) = \frac{-1 + \csc h^2(\zeta)}{2\csc h(\zeta)}, \ \cosh w(\zeta) = \frac{1 + \csc h^2(\zeta)}{2\csc h(\zeta)}. \quad (22)$$

## 3 Solutions of the elliptic-like equation

Considering the elliptic-like equation [13]

$$A\psi''(\zeta) + B\psi(\zeta) + D\psi^3(\zeta) = 0, \quad (23)$$

where $A$, $B$, $D$ are arbitrary constants.

### 3.1 Application of the power-exponential function method

We assume that Eq (23) has the following formal solution

$$\psi(\zeta) = \frac{Me^{\lambda\zeta} + S}{Re^{2\lambda\zeta} + Ne^{\lambda\zeta} + T}, \quad (24)$$

where $M$, $S$, $R$, $N$, $T$ and $\lambda$ are constants to be determined.

Substituting Eq (24) into Eq (23) and setting the coefficients of all powers of $e^{j\lambda\zeta}$ ($j = 0, 1, \cdots, 5$) to zero, we have

$$\begin{cases} BMR^2 + A\lambda^2 MR^2 = 0, \\ BSR^2 + 4A\lambda^2 SR^2 - 4\lambda^2 MRN + 2BMRN = 0, \\ BMN^2 + DM^3 - 6A\lambda^2 MRT + 3A\lambda^2 SRN + 2BSRN + 2BMRT = 0, \\ DSN^2 - 4A\lambda^2 SRT + 2BSRT + 3DM^2 S + A\lambda^2 SN^2 - A\lambda^2 MNT + 2BMNT = 0, \\ BMT^2 + 3DMS^2 + A\lambda^2 MT^2 - A\lambda^2 SNT + 2BSNT = 0, \\ DS^3 + BST^2 = 0. \end{cases} \tag{25}$$

Solving the algebraic Eq (25) with the aid of Maple, we get the following solution

$$R = -\frac{DM^2}{8BT}, \quad T = T, \quad N = 0, \quad A\lambda^2 + B = 0, \quad S = 0, \quad M = M. \tag{26}$$

Substituting Eq (26) into Eq (24), we can obtain many kinds of solutions of Eq (23). If $A\lambda^2 + B = 0$, $S = 0$ and $M$, $T$ are arbitrary constants, Eq (23) has a rational-type solution

$$\psi_1(\zeta) = \frac{-8MBTe^{\lambda\zeta}}{DM^2 e^{2\lambda\zeta} - 8BT^2} \tag{27}$$

If $DM^2 + 8BT^2 = 0$, Eq (23) has a bell profile solution

$$\psi_2(\zeta) = -4MBT\, sech(\lambda\zeta). \tag{28}$$

If $DM^2 - 8BT^2 = 0$, Eq (23) has a singular solution

$$\psi_3(\zeta) = -4MBT\, csch(\lambda\zeta). \tag{29}$$

### 3.2 Application of the extended hyperbolic auxiliary equation method

Considering the homogeneous balance between $\psi''(\zeta)$ and $\psi^3(\zeta)$ in Eq (23), we suppose that the solution of Eq (23) can be expressed by

$$\psi(\zeta) = a_0 + a_1 sinhw(\zeta) + b_1 coshw(\zeta), \tag{30}$$

where $a_0$, $a_1$, $b_1$ are constants to be determined, $sinhw(\zeta)$ and $coshw(\zeta)$ satisfy Eq (8).

Substituting Eq (30) along with Eq (8) into Eq (23) and collecting the coefficients of $\sinh^p w$
$(\zeta) \cosh^r w(\zeta)$ $(r = 0, 1; p = 0, 1, 2, 3)$, we have

$$
\begin{cases}
Ba_0 + D(a_0^3 + 3a_0 b_1^3) = 0, \\[2mm]
A\left(a_1 f + \dfrac{3b_1 g}{2} + 2a_1 h\right) + Ba_1 + 3D(a_0^2 a_1 + a_1 b_1^2) = 0, \\[2mm]
A\left(b_1 f + \dfrac{a_1 g}{2} + b_1 h\right) + Bb_1 + D(3a_0^2 b_1 + b_1^3) = 0, \\[2mm]
6Da_0 a_1 b_1 = 0, \\[2mm]
3D(a_0 a_1^2 + a_0 b_1^2) = 0, \\[2mm]
2A(a_1 g + b_1 h) + D(3a_1^2 b_1 + b_1^3) = 0, \\[2mm]
2A(b_1 g + a_1 h) + D(a_1^3 + 3a_1 b_1^2) = 0.
\end{cases}
\tag{31}
$$

Solving the algebraic Eq (31) with the aid of Maple, we get the following solutions.
**Family 1** $g \neq 0$

$$
a_1 = \frac{\sqrt{6D(\Delta + N)}}{6D}, \quad b_1 = \frac{(Af + B)\sqrt{6D(\Delta + N)}}{3DAg} - \frac{[6D(\Delta + N)]^{\frac{3}{2}}}{108D^2 Ag}, \quad a_0 = 0.
\tag{32}
$$

$$
a_1 = -\frac{\sqrt{6}\sqrt{D(\Delta + N)}}{6D}, b_1 = -\frac{(Af + B)\sqrt{6D(\Delta + N)}}{3DAg} + \frac{[6D(\Delta + N)]^{\frac{3}{2}}}{108D^2 Ag}, a_0 = 0.
\tag{33}
$$

$$
a_1 = \frac{\sqrt{6}\sqrt{D(-\Delta + N)}}{6D}, b_1 = \frac{(Af + B)\sqrt{6D(-\Delta + N)}}{3DAg} - \frac{[6D(-\Delta + N)]^{\frac{3}{2}}}{108D^2 Ag}, a_0 = 0.
\tag{34}
$$

$$
a_1 = -\frac{\sqrt{6}\sqrt{D(-\Delta + N)}}{6D}, b_1 = -\frac{(Af + B)\sqrt{6D(-\Delta + N)}}{3DAg} + \frac{[6D(-\Delta + N)]^{\frac{3}{2}}}{108D^2 Ag}, a_0 = 0
\tag{35}
$$

where

$$
\triangle = \sqrt{4A^2 f^2 + 8AfB + 4A^2 fh + 4B^2 + 4BAh + A^2 h^2 + 9A^2 g^2}, N = 4Af + 4B - Ah.
$$

**Family 2** $g = 0$

$$
a_1 = \pm\sqrt{\frac{Af + B}{D}}, \quad a_0 = b_1 = 0.
\tag{36}
$$

Substituting Eq (32) into Eq (30) along with Eqs (9)–(22), we can obtain many kinds of
dark solitary wave, bell profile solitary wave and Jacobian elliptic function solutions of Eq (23),
which are listed below.
(i) Seven Jacobi elliptic doubly periodic-type solutions

$$
\psi_4(\zeta) = \frac{2HJ - HIcn^2(\zeta)}{2sn(\zeta)},
\tag{37}
$$

where

$$H = \frac{\sqrt{3D(\sqrt{B^2 + 10A^2 - 2AB - 2A(B + 8A)m^2 + 10A^2m^4} + 2B - 5Am^2 - 5A)}}{9AD(m^2 - 1)},$$

$$I = B - 4A + 2Am^2 - \sqrt{B^2 + 10A^2 - 2AB - 2A(B + 8A)m^2 + 10A^2m^4},$$

$$J = A - B + Am^2 + \sqrt{B^2 + 10A^2 - 2AB - 2A(B + 8A)m^2 + 10A^2m^4}.$$

and

$$\psi_5(\zeta) = \frac{2HJ - HIsn^2(\zeta)}{2cn(\zeta)}, \tag{38}$$

where

$$H = \frac{\sqrt{3D(\sqrt{B^2 + 10A^2 - 2AB + 4A(B - A)m^2 + 4A^2m^4} + 2B + 10Am^2 - 5A)}}{9AD},$$

$$I = 4A - B - 2Am^2 + \sqrt{B^2 + 10A^2 - 2AB + 4A(B - A)m^2 + 4A^2m^4},$$

$$J = B - A + 2Am^2 - \sqrt{B^2 + 10A^2 - 2AB + 4A(B - A)m^2 + 4A^2m^4}.$$

and

$$\psi_6(\zeta) = \frac{2HJ - HIm^2[1 + cn^2(\zeta)]}{2dn(\zeta)}, \tag{39}$$

where

$$H = \frac{\sqrt{3D(\sqrt{B^2 + 4A^2 + 4AB - 2A(B + 2A)m^2 + 10A^2m^4} + 2B - 5Am^2 + 10A)}}{9ADm^2},$$

$$I = 4Am^2 - 2A - B + \sqrt{B^2 + 4A^2 + 4AB - 2A(B + 2A)m^2 + 10A^2m^4},$$

$$J = Am^2 - B - 2A + \sqrt{B^2 + 4A^2 + 4AB - 2A(B + 2A)m^2 + 10A^2m^4}.$$

and

$$\psi_7(\zeta) = \frac{HJ + HIsc^2(\zeta)}{2sc(\zeta)}, \tag{40}$$

where

$$H = \frac{\sqrt{3D(\sqrt{B^2 + 4A^2 + 4AB - 2A(B + 2A)m^2 + 10A^2m^4} + 2B + Am^2 - 2A)}}{9ADm^2},$$

$$I = 4Am^2 - B - 2A + \sqrt{B^2 + 4A^2 + 4AB - 2A(B + 2A)m^2 + 10A^2m^4},$$

$$J = \sqrt{B^2 + 4A^2 + 4AB - 2A(B + 2A)m^2 + 10A^2m^4} - B - 2A - 2Am^2.$$

and

$$\psi_8(\zeta) = \frac{HJ + HIsd^2(\zeta)}{2sd(\zeta)}, \tag{41}$$

where

$$H = \frac{\sqrt{3D(\Delta + 2B - 5A + 7Am^2 - 3Am^4)}}{9AD(m^4 - m^2 - 1)},$$

$$I = 3Am^4 - Am^2 - 4A + B - \Delta, J = 5Am^4 - 3Am^2 + 2A + B - \Delta,$$

$$\Delta = \sqrt{9A^2m^8 - 18A^2m^6 - 5A^2m^4 + 2A(2B + 7A)m^2 + B^2 - 2AB + 10A^2}.$$

and

$$\psi_9(\zeta) = \frac{J + Hcd^2(\zeta)}{2cd(\zeta)}, \tag{42}$$

where

$$J = \frac{\sqrt{3D(\sqrt{B^2 - 2AB + 10A^2} + 2B - 5A)(B - 4A - \sqrt{B^2 - 2AB + 10A^2})}}{9AD},$$

$$H = \frac{\sqrt{3D(\sqrt{B^2 - 2AB + 10A^2} + 2B - 5A)(2A + B - \sqrt{B^2 - 2AB + 10A^2})}}{9AD}.$$

and

$$\psi_{10}(\zeta) = \frac{HJ[1 \pm dn(\zeta)]}{2sn(\zeta)} + \frac{HIsn(\zeta)}{2[1 \pm dn(\zeta)]}, \tag{43}$$

where

$$J = 2Am^2 - 3Am^4 + 4B - A - \triangle, I = 3Am^4 + 2Am^2 + 4B - 7A - \triangle,$$

$$H = \frac{\sqrt{3D(\sqrt{16B^2 - 32AB + 25A^2 + 16A(B - A)m^2 - 14A^2m^4 + 9A^2m^8} + 8B - 11A - 3Am^4 + 4Am^2)}}{18AD(m^4 - 1)},$$

$$\triangle = \sqrt{16B^2 - 32AB + 25A^2 + 16A(B - A)m^2 - 14A^2m^4 + 9A^2m^8}.$$

(ii) A bell profile solitary wave solution

$$\psi_{11}(\zeta) = \frac{1}{2}[J + H \sec h^2(\zeta)] \cosh(\zeta), \tag{44}$$

where

$$J = \frac{\sqrt{3D(\sqrt{B^2 + 2AB + 10A^2} + 2B + 5A)(\sqrt{B^2 + 2AB + 10A^2} - 4A - B)}}{9AD},$$

$$H = \frac{\sqrt{3D(\sqrt{B^2 + 2AB + 10A^2} + 2B + 5A)(2A - B + \sqrt{B^2 + 2AB + 10A^2})}}{9AD}.$$

(iii) A singular soliton solution

$$\psi_{12}(\zeta) = \frac{J + H \csc h^2(\zeta)}{2 \csc h(\zeta)}, \tag{45}$$

where

$$J = \frac{\sqrt{3D(\sqrt{B^2 + 2AB + 10A^2} + 2B - A)(B - 2A - \sqrt{B^2 + 2AB + 10A^2})}}{9AD},$$

$$H = \frac{\sqrt{3D(\sqrt{B^2 + 2AB + 10A^2} + 2B - A)(B + 4A - \sqrt{B^2 + 2AB + 10A^2})}}{9AD}.$$

(iv) Four Jacobi elliptic doubly periodic-type solutions

$$\psi_{13}(\zeta) = \pm \sqrt{\left| \frac{m^2 A + B}{D} \right|} \frac{sn(\zeta)}{cn(\zeta)}. \tag{46}$$

and

$$\psi_{14}(\zeta) = \pm \sqrt{\left| \frac{A + B}{D} \right|} \frac{msn(\zeta)}{dn(\zeta)}. \tag{47}$$

and

$$\psi_{15}(\zeta) = \pm \sqrt{\left| \frac{B - A}{D} \right|} \left[ \frac{1 \pm dn(\zeta)}{2msn(\zeta)} + \frac{msn(\zeta)}{2[1 \pm dn(\zeta)]} \right]. \tag{48}$$

and

$$\psi_{16}(\zeta) = \pm \sqrt{\left| \frac{B - m^2 A}{D} \right|} \frac{cn(\zeta)}{sn(\zeta)}. \tag{49}$$

(v) A dark soliton wave solution

$$\psi_{17}(\zeta) = \pm \sqrt{\left| \frac{B - 4A}{D} \right|} \frac{(\tanh^2(\zeta) - 1)}{2\tanh(\zeta)}. \tag{50}$$

## 4 Applications to the Hirota equation

In this section, we mainly consider the defocusing Hirota Eq (4)

$$iq_t + \frac{1}{2}q_{xx} - |q|^2 q + i\alpha(-q_{xxx} + 6|q|^2 q_x) = 0, \tag{51}$$

where $\alpha \in R$.

This equation was introduced by Hirota and exact envelope-soliton solutions were established. One can see that in one limit of $\alpha = 0$, the equation reduces to the nonlinear Schrödinger equation which describes plane self-focusing and one-dimensional self-modulation of waves in nonlinear dispersive media. Eq (51) has been studied in some literatures [14–16]. Here, we will find some new solutions of Eq (51) by the method proposed in this paper.

Considering the transformation

$$q(x,t) = \psi(\zeta)exp[i(kx - \omega t)] , \quad \zeta = -\lambda x + t + zeta_0, \tag{52}$$

where $k$, $\omega$ and $\lambda$ are constants to be determined, $\zeta_0$ is an arbitrary constant.

Substituting Eq (52) into Eq (51), we have

$$(\lambda^2 - 3\alpha\lambda^2 k)\phi'' + (\omega - k^2 + \alpha k^3)\phi + (2 - 6\alpha k)\phi^3 = 0, \tag{53}$$

$$\alpha\lambda^3\phi''' - (1 - 2k\lambda + 3\alpha k^2\lambda)\phi' + 6\alpha\lambda\phi^2\phi' = 0, \tag{54}$$

under the constraint condition

$$\omega = \frac{8\alpha k^2\lambda(\alpha k - 1) + k(3\alpha + 2\lambda) - 1}{\alpha\lambda}. \tag{55}$$

Then (53) and (54) become

$$A\psi''(\zeta) + B\psi(\zeta) + D\psi^3(\zeta) = 0, \tag{56}$$

Eq (56) coincides with (23), where

$$A = 1, \qquad B = \frac{k\lambda(2 - 3\alpha k) - 1}{\alpha\lambda^3}, \qquad D = \frac{2}{\lambda^2}.$$

Then the solutions of (51) are

$$q(x,t) = \psi(\zeta)exp[i(kx - \omega t)] , \quad \zeta = -\lambda x + t + \zeta_0, \tag{57}$$

where $\psi(\zeta)$ appearing in these solutions are given by (27–29) and (37)–(50).

Substituting Eqs (27)–(29) and (37)–(50) into Eq (57), we obtain the following dark solitary wave, bell profile solitary wave and Jacobian elliptic function solutions of the Hirota Eq (51).

$$q_1(x,t) = \left[\frac{8MBTe^{\lambda\zeta}}{8BT^2 - DM^2 e^{2\lambda\zeta}}\right] exp[i(kx - \omega t)].$$

$$q_2(x,t) = -4MBTsech(\lambda\zeta)exp[i(kx - \omega t)].$$

$$q_3(x,t) = -4MBTcsch(\lambda\zeta)exp[i(kx - \omega t)].$$

$$q_4(x,t) = \left[\frac{2HJ - HIcn^2(\zeta)}{2sn(\zeta)}\right] exp[i(kx - \omega t)],$$

where

$$H = \frac{\sqrt{3D(\sqrt{B^2 + 10A^2 - 2AB - 2A(B + 8A)m^2 + 10A^2m^4} + 2B - 5Am^2 - 5A)}}{9AD(m^2 - 1)},$$

$$I = B - 4A + 2Am^2 - \sqrt{B^2 + 10A^2 - 2AB - 2A(B + 8A)m^2 + 10A^2m^4},$$

$$J = A - B + Am^2 + \sqrt{B^2 + 10A^2 - 2AB - 2A(B + 8A)m^2 + 10A^2m^4}.$$

$$q_5(x, t) = \left[ \frac{2HJ - HIsn^2(\zeta)}{2cn(\zeta)} \right] exp[i(kx - \omega t)],$$

where

$$H = \frac{\sqrt{3D(\sqrt{B^2 + 10A^2 - 2AB + 4A(B - A)m^2 + 4A^2m^4} + 2B + 10Am^2 - 5A)}}{9AD},$$

$$I = 4A - B - 2Am^2 + \sqrt{B^2 + 10A^2 - 2AB + 4A(B - A)m^2 + 4A^2m^4},$$

$$J = B - A + 2Am^2 - \sqrt{B^2 + 10A^2 - 2AB + 4A(B - A)m^2 + 4A^2m^4}.$$

$$q_6(x, t) = \left\{ \frac{2HJ - HIm^2[1 + cn^2(\zeta)]}{2dn(\zeta)} \right\} exp[i(kx - \omega t)],$$

where

$$H = \frac{\sqrt{3D(\sqrt{B^2 + 4A^2 + 4AB - 2A(B + 2A)m^2 + 10A^2m^4} + 2B - 5Am^2 + 10A)}}{9ADm^2},$$

$$I = 4Am^2 - 2A - B + \sqrt{B^2 + 4A^2 + 4AB - 2A(B + 2A)m^2 + 10A^2m^4},$$

$$J = Am^2 - B - 2A + \sqrt{B^2 + 4A^2 + 4AB - 2A(B + 2A)m^2 + 10A^2m^4}.$$

$$q_7(x, t) = \left[ \frac{HJ + HIsc^2(\zeta)}{2sc(\zeta)} \right] exp[i(kx - \omega t)],$$

where

$$H = \frac{\sqrt{3D(\sqrt{B^2 + 4A^2 + 4AB - 2A(B + 2A)m^2 + 10A^2m^4} + 2B + Am^2 - 2A)}}{9ADm^2},$$

$$I = 4Am^2 - B - 2A + \sqrt{B^2 + 4A^2 + 4AB - 2A(B + 2A)m^2 + 10A^2m^4},$$

$$J = \sqrt{B^2 + 4A^2 + 4AB - 2A(B + 2A)m^2 + 10A^2m^4} - B - 2A - 2Am^2.$$

$$q_8(x, t) = \left[ \frac{HJ + HIsd^2(\zeta)}{2sd(\zeta)} \right] exp[i(kx - \omega t)],$$

where

$$H = \frac{\sqrt{3D(\Delta + 2B - 5A + 7Am^2 - 3Am^4)}}{9AD(m^4 - m^2 - 1)},$$

$$I = 3Am^4 - Am^2 - 4A + B - \Delta, \quad J = 5Am^4 - 3Am^2 + 2A + B - \Delta,$$

$$\Delta = \sqrt{9A^2m^8 - 18A^2m^6 - 5A^2m^4 + 2A(2B + 7A)m^2 + B^2 - 2AB + 10A^2}.$$

$$q_9(x, t) = \left[ \frac{J + Hcd^2(\zeta)}{2cd(\zeta)} \right] exp[i(kx - \omega t)],$$

where

$$J = \frac{\sqrt{3D(\sqrt{B^2 - 2AB + 10A^2} + 2B - 5A)(B - 4A - \sqrt{B^2 - 2AB + 10A^2})}}{9AD},$$

$$H = \frac{\sqrt{3D(\sqrt{B^2 - 2AB + 10A^2} + 2B - 5A)(2A + B - \sqrt{B^2 - 2AB + 10A^2})}}{9AD}.$$

$$q_{10}(x, t) = \left\{ \frac{HJ[1 \pm dn(\zeta)]}{2sn(\zeta)} + \frac{HIsn(\zeta)}{2[1 \pm dn(\zeta)]} \right\} exp[i(kx - \omega t)],$$

where

$$J = 2Am^2 - 3Am^4 + 4B - A - \triangle, I = 3Am^4 + 2Am^2 + 4B - 7A - \triangle,$$

$$H = \frac{\sqrt{3D(\sqrt{16B^2 - 32AB + 25A^2 + 16A(B - A)m^2 - 14A^2m^4 + 9A^2m^8} + 8B - 11A - 3Am^4 + 4Am^2)}}{18AD(m^4 - 1)},$$

$$\triangle = \sqrt{16B^2 - 32AB + 25A^2 + 16A(B - A)m^2 - 14A^2m^4 + 9A^2m^8}.$$

$$q_{11}(x, t) = \frac{1}{2}[J + H \sec h^2(\zeta)] \cosh(\zeta) exp[i(kx - \omega t)].$$

where

$$J = \frac{\sqrt{3D(\sqrt{B^2 + 2AB + 10A^2} + 2B + 5A)(\sqrt{B^2 + 2AB + 10A^2} - 4A - B)}}{9AD},$$

$$H = \frac{\sqrt{3D(\sqrt{B^2 + 2AB + 10A^2} + 2B + 5A)(2A - B + \sqrt{B^2 + 2AB + 10A^2})}}{9AD}.$$

$$q_{12}(x, t) = \left[ \frac{J + H \csc h^2(\zeta)}{2 \csc h(\zeta)} \right] exp[i(kx - \omega t)],$$

where

$$J = \frac{\sqrt{3D(\sqrt{B^2 + 2AB + 10A^2} + 2B - A)(B - 2A - \sqrt{B^2 + 2AB + 10A^2})}}{9AD},$$

$$H = \frac{\sqrt{3D(\sqrt{B^2 + 2AB + 10A^2} + 2B - A)(B + 4A - \sqrt{B^2 + 2AB + 10A^2})}}{9AD}.$$

$$q_{13}(x, t) = \pm\sqrt{\left|\frac{m^2A + B}{D}\right|}\left[\frac{sn(\zeta)}{cn(\zeta)}\right]exp[i(kx - \omega t)].$$

$$q_{14}(x, t) = \pm\sqrt{\left|\frac{A + B}{D}\right|}\left[\frac{msn(\zeta)}{dn(\zeta)}\right]exp[i(kx - \omega t)].$$

$$q_{15}(x, t) = \pm\sqrt{\left|\frac{B - A}{D}\right|}\left[\frac{1 \pm dn(\zeta)}{2msn(\zeta)} + \frac{msn(\zeta)}{2[1 \pm dn(\zeta)]}\right]exp[i(kx - \omega t)].$$

$$q_{16}(x, t) = \pm\sqrt{\left|\frac{B - m^2A}{D}\right|}\left[\frac{cn(\zeta)}{sn(\zeta)}\right]exp[i(kx - \omega t)].$$

$$q_{17}(x, t) = \pm\sqrt{\left|\frac{B - 4A}{D}\right|}\left[\frac{\tanh^2(\zeta) - 1}{2\tanh(\zeta)}\right]exp[i(kx - \omega t)].$$

**Remark** This approach can also be applied to solve the other nonlinear PDEs. For certain significant PDEs in the fields of Mathematics and Physics, if they can be converted into the form of elliptic Eq (23), then their solutions are readily obtained. In addition, to the best of our knowledge, the solutions $q_1$, $q_4 - q_{10}$ and $q_{15}$ of Eq (51) are new, which can not be found in [14–16]. In addition, since Eqs (33)–(35) are similar to Eq (32), here we only list the solutions obtained by substituting Eqs (32) and (36) into Eq (23) along with Eqs (9)–(22). In addition, if the modulus $m \to 0$, some trigonometric function solutions can be obtained where some solitary wave solutions can also be obtained if the modulus $m \to 1$. Here we omit them.

This section investigates the physical interpretations of the above-constructed solutions Eq (51) that have been demonstrated in some distinct graphs. Fig 1 illustrates the solitary wave solution $q_{10}$ of Eq (51) with the integration constant be one and $m = 1/2$ at times $t = 3.14$. Fig 2 illustrates the solitary wave solution $q_{15}$ of Eq (51) with the integration constant be one and $m = 1/2$ at times $t = 3.14$. These figures explain the nature of solitary wave solution infulenced by mass's location and time dependency and the many representations of the solitary wave solution in polar, three-dimensional graphs. These two figures represent bell profile solitary wave solutions and Jacobian elliptic function solutions in some different graphs'types. These apply to water waves and many other fields like plasma and atmosphere physics, Bose-Einstein condenses, nonlinear optics, superconductivity, and so on.

## 5 Conclusions

In this paper, we have utilized the subsidiary elliptic-like equation, the power-exponential function method and the extended hyperbolic auxiliary equation method to study the Hirota

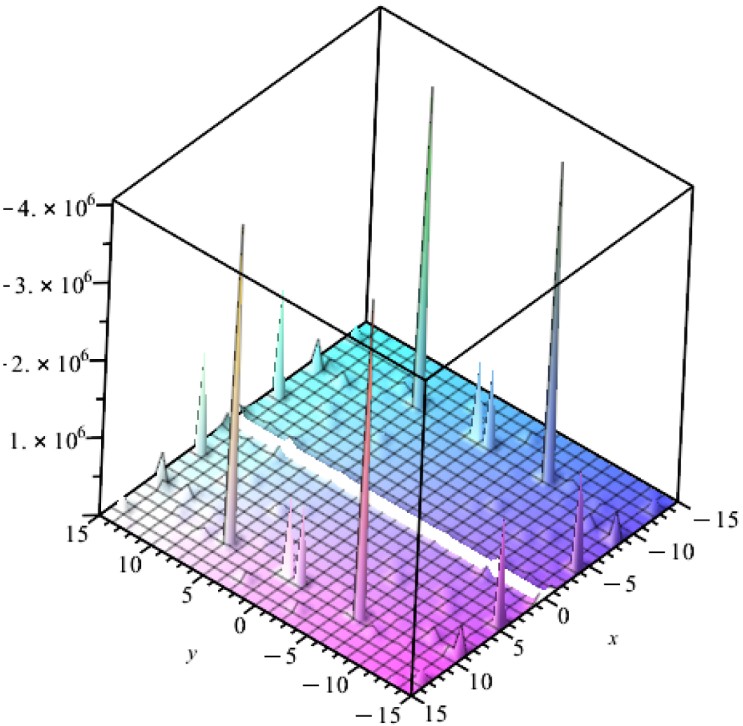

**Fig 1. Solitary ware solution q10 of Eq (51), with the integration constant be one, and m = 1/2 at times t = 3.14.**

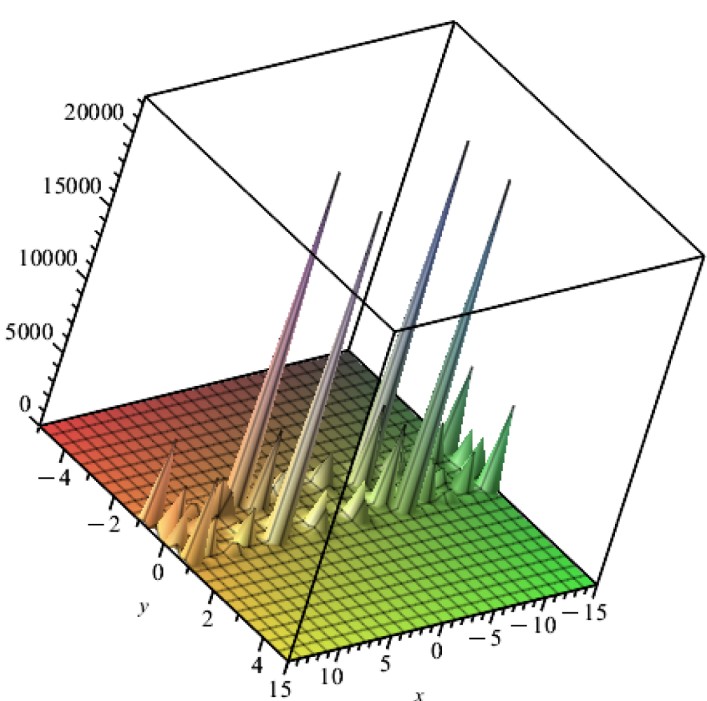

**Fig 2. Solitary wave solution q15 of Eq (51), with the integration constant be one, and m = l/2 at times t = 3.14.**

equation. As a result, a series of solutions including the combined nondegenerative Jacobi elliptic function solutions, degenerative solutions, soliton-like solution, singular soliton-like solution, combined bell profile solution and kind profile solution are derived. Some of these solutions are found for the first time. The solutions obtained may be of important significance for the explanation of some practical physical problem. This is a new application of the subsidiary elliptic-like equation, the power-exponential function method and the extended hyperbolic auxiliary equation method. This approach can also be applied to solve the other nonlinear partial differential equations.

The limitations of the study lie in that the method is only used for integer-order equations, potential applications to fractional derivative equations will be also interesting for practical problems [17, 18]. This is under the scope of our future investigation.

## Author Contributions

**Conceptualization:** Fuzhang Wang, Sohail Nadeem.

**Data curation:** Bacui Li.

**Formal analysis:** Bacui Li.

**Funding acquisition:** Fuzhang Wang.

**Investigation:** Bacui Li.

**Methodology:** Bacui Li, Fuzhang Wang, Sohail Nadeem.

**Software:** Bacui Li.

**Supervision:** Fuzhang Wang, Sohail Nadeem.

**Validation:** Bacui Li.

**Writing – original draft:** Bacui Li, Fuzhang Wang.

**Writing – review & editing:** Sohail Nadeem.

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
