## [Decision Letter · Decision Letter 0]

9 Feb 2024

PONE-D-24-02488Explicit and exact travelling wave solutions for Hirota equation and Computerized MechanizationPLOS ONE

Dear Dr. Wang,

Thank you for submitting your manuscript to PLOS ONE. After careful consideration, we feel that it has merit but does not fully meet PLOS ONE’s publication criteria as it currently stands. Therefore, we invite you to submit a revised version of the manuscript that addresses the points raised during the review process.

We look forward to receiving your revised manuscript.

Kind regards,

Tao Peng

Academic Editor

PLOS ONE

Journal Requirements:

"The work was supported by the Natural Science Foundation of Jiangxi Province (Project No. 20224BAB201018)."

"The work was supported by the Natural Science Foundation of Jiangxi Province (Project No. 20224BAB201018)."

 [copy in funding statement].     

4. We note that your Data Availability Statement is currently as follows: "All relevant data are within the manuscript and its Supporting Information files."

Additional Editor Comments: 

Based on the rigorous review procedure, the Major Revision is decided. The authors should point-to-point response the Reviewers’ comments, and it can be considered to publish after the careful revision. In addition, the authors should complete this rebuttal process within the rebuttal deadline, and if you need more time for this rebuttal process on the reasonable request, please contact us. Submission with both clean version and highlight version of the revised manuscript is recommended, which can help Editor board to make the quick decision, but not a mandatory requirement.

Reviewers' comments:

Reviewer's Responses to Questions

**Comments to the Author**

1. Is the manuscript technically sound, and do the data support the conclusions?

Reviewer #1: Yes

Reviewer #2: Yes

2. Has the statistical analysis been performed appropriately and rigorously? 

Reviewer #1: Yes

Reviewer #2: Yes

3. Have the authors made all data underlying the findings in their manuscript fully available?

Reviewer #1: Yes

Reviewer #2: Yes

4. Is the manuscript presented in an intelligible fashion and written in standard English?

Reviewer #1: Yes

Reviewer #2: Yes

5. Review Comments to the Author

Reviewer #1: In the study, innovative methods were used to derive new solutions for the Hirota equation, crucial for modeling nonlinear phenomena. The findings include solitary wave and Jacobian elliptic function solutions. The approach, applicable to other nonlinear PDEs, emphasizes the potential for broader implications in mathematics and physics. Further exploration and application of these methods could yield significant advancements in understanding complex systems. I would suggest accepting it after the following major concerns are addressed:

1. In the introduction part, integrating a brief overview of the methods' novelty compared to traditional approaches can captivate the reader's interest and underscore the study's significance in the broader field of nonlinear optics.

2. Add the limitations of the study in the conclusion part. Discussing potential applications or implications in practical scenarios would bridge the gap between theoretical solutions and real-world relevance.

3. In "Introduction of the extended hyperbolic auxiliary equation method" part, to improve this section, consider incorporating a more detailed explanation of the physical significance and applicability of each case scenario.

4. To enhance "Solutions of the elliptic-like equation" section, incorporating a clear, concise summary of each solution's physical interpretation could be beneficial. Visual representations or graphs of the solutions could aid in comprehensibility.

5. Improving the article's structure: could involve organizing content into clearly defined sections.

Reviewer #2: Review for PONE-D-24-02488

The research work focuses on finding explicit and exact travelling wave solutions for the Hirota equation through various mathematical methods. The paper presents a significant contribution by introducing new solutions for the Hirota equation, enhancing the understanding of nonlinear physical phenomena. Here are some suggestions for this article:

1. How do the chosen methods (power-exponential function method and the extended hyperbolic auxiliary equation method) compare in efficiency and accuracy with other contemporary approaches to solving the Hirota equation?

2. Can the authors elaborate on the unique aspects of the solutions found for the Hirota equation and how these contribute to advancing the field?

3. What are the theoretical implications of the new solutions for understanding complex systems modeled by the Hirota equation?

4. Are there specific practical applications in physics or engineering where the new solutions could be particularly beneficial? How do these solutions improve upon existing models or methodologies in practical scenarios?

5. What are the limitations of the current study, and what assumptions have been made in the application of the methods? How do these limitations affect the interpretation of the results?

6. Based on the findings, what future research directions do the authors suggest? Are there any particular areas within the field where these solutions could open up new lines of inquiry?

7. Could the authors provide more detail on the data or simulations used to verify the solutions, ensuring reproducibility for other researchers?

8. How do the solutions derived in this study perform in comparison to existing solutions to the Hirota equation in terms of stability, solvability, and physical relevance?

Overall, the paper makes contributions to the field of nonlinear partial differential equations, particularly in the study of the Hirota equation. Expanding on certain aspects as suggested could make the paper even more robust and impactful.

6. PLOS authors have the option to publish the peer review history of their article (what does this mean?). If published, this will include your full peer review and any attached files.

Reviewer #1: **Yes: **Taolin Qin

Reviewer #2: No

---

## [Author Response · Author response to Decision Letter 0]

13 Mar 2024

We highly appreciate your detailed and helpful comments on our manuscript, which have helped to significantly improve academic quality and readability of this paper. The revised version has considered all of your recommendations and criticisms.

Reply to reviewer 1: 

1. In the introduction part, integrating a brief overview of the methods' novelty compared to traditional approaches can captivate the reader's interest and underscore the study's significance in the broader field of nonlinear optics.

We have added a brief overview of the current method and traditional method. In short, the key idea of the traditional auxiliary equation method is to use the solutions of new first-order nonlinear ordinary differential equation instead of tanh(.) in tanh-function method and extended tanh-function method. By using the extended hyperbolic auxiliary equation method, a series of new travelling wave solutions have been obtained including not only Jacobian elliptic function solutions but also solitary wave solutions and trigonometric function solutions.

2. Add the limitations of the study in the conclusion part. Discussing potential applications or implications in practical scenarios would bridge the gap between theoretical solutions and real-world relevance.

According to your suggestion, we have added the limitations of the study in the conclusion part. In short, the limitations of the study lie in that the method is only used for integer-order equations, potential applications to fractional derivative equations will be also interesting for practical problems. 

3. In "Introduction of the extended hyperbolic auxiliary equation method" part, to improve this section, consider incorporating a more detailed explanation of the physical significance and applicability of each case scenario.

According to your suggestion, we have provided a more detailed explanation of the physical significance and applicability of each case scenario in "Introduction of the extended hyperbolic auxiliary equation method" part. 

4. To enhance "Solutions of the elliptic-like equation" section, incorporating a clear, concise summary of each solution's physical interpretation could be beneficial. Visual representations or graphs of the solutions could aid in comprehensibility.

According to your suggestion, we have provided a clear and concise summary of the physical explanations for each solution in "Solutions of the elliptic-like equation" section. In addition, we have drawn some pictures and added corresponding description.

5. Improving the article's structure: could involve organizing content into clearly defined sections.

According to your suggestion, we have re-organized Section 2 into clearly defined sub-sections to improve the article's structure.

Reply to reviewer 2: 

1.How do the chosen methods (power-exponential function method and the extended hyperbolic auxiliary equation method) compare in efficiency and accuracy with other contemporary approaches to solving the Hirota equation?

In this paper, we have investigated the explicit and exact travelling wave solutions of the Hirota equation. Compared to the traditional auxiliary equation method, a series of new travelling wave solutions have been obtained by using the extended hyperbolic auxiliary equation method which include not only Jacobian elliptic function solutions but also solitary wave solutions and trigonometric function solutions.

2.Can the authors elaborate on the unique aspects of the solutions found for the Hirota equation and how these contribute to advancing the field?

The Hirota equation is a very important PDE, which is commonly used to describe the mathematical model of optical soliton propagation in dispersive optical fibers from the field of nonlinear optics. The solutions obtained in the paper are useful for further understanding the propagation of optical fibers in nonlinear optical fibers. Moreover, the results of this article can also make up for the lack of qualitative analysis of Hirota equation and its perturbation system.

3.What are the theoretical implications of the new solutions for understanding complex systems modeled by the Hirota equation?

Exact solutions extracted from Hirota equation are known to provide complete information on real-world occurrences. These solutions obtained in this manuscript are believed to play a role in understanding the dynamical aspects of the Hirota equation. 

4.Are there specific practical applications in physics or engineering where the new solutions could be particularly beneficial? How do these solutions improve upon existing models or methodologies in practical scenarios?

Your question is very interesting. The main goal of our manuscript is to we present the explicit solutions of the subsidiary elliptic-like equation. With the aid of the subsidiary elliptic-like equation and a simple transformation, we can derive the exact solutions of Hirota equation. The practical application in physics or engineering beyond the scope of our research, we will consider it as our future investigation. 

5.What are the limitations of the current study, and what assumptions have been made in the application of the methods? How do these limitations affect the interpretation of the results?

According to your suggestion, we have added the limitations of the study in the conclusion part. In short, the limitations of the study lie in that the method is only used for integer-order equations.

6.Based on the findings, what future research directions do the authors suggest? Are there any particular areas within the field where these solutions could open up new lines of inquiry?

In the conclusion part, we have added potential research directions which include the applications to fractional derivative equations. There is a wide range of applications, part of which is being under investigation by the authors recently.

7.Could the authors provide more detail on the data or simulations used to verify the solutions, ensuring reproducibility for other researchers?

To ensure reproducibility for other researchers, we have tried our best to make the whole manuscript clear-to-understand. Researchers who are interested in our work can reproduce our result easily with the help of MAPLE.

8.How do the solutions derived in this study perform in comparison to existing solutions to the Hirota equation in terms of stability, solvability, and physical relevance?

Compared with the traditional auxiliary equation method, a series of new travelling wave solutions have been obtained by using the extended hyperbolic auxiliary equation method which include the Jacobian elliptic function solutions, solitary wave solutions and trigonometric function solutions.

---

## [Decision Letter · Decision Letter 1]

29 Apr 2024

PONE-D-24-02488R1Explicit and exact travelling wave solutions for Hirota equation and Computerized MechanizationPLOS ONE

Dear Dr. Wang,

Thank you for submitting your manuscript to PLOS ONE. After careful consideration, we feel that it has merit but does not fully meet PLOS ONE’s publication criteria as it currently stands. Therefore, we invite you to submit a revised version of the manuscript that addresses the points raised during the review process.

We look forward to receiving your revised manuscript.

Kind regards,

Tao Peng

Academic Editor

PLOS ONE

Journal Requirements:

**Additional Editor Comments:**

Based on the careful revision, this work could be considered to be accepted after the minor revision, where it still contains several issues, 1) language should be polished by the English native speaker, 2) image and math equations should be polished.

Reviewers' comments:

Reviewer's Responses to Questions

**Comments to the Author**

1. If the authors have adequately addressed your comments raised in a previous round of review and you feel that this manuscript is now acceptable for publication, you may indicate that here to bypass the “Comments to the Author” section, enter your conflict of interest statement in the “Confidential to Editor” section, and submit your "Accept" recommendation.

Reviewer #1: All comments have been addressed

Reviewer #2: All comments have been addressed

2. Is the manuscript technically sound, and do the data support the conclusions?

Reviewer #1: Yes

Reviewer #2: Yes

3. Has the statistical analysis been performed appropriately and rigorously? 

Reviewer #1: Yes

Reviewer #2: Yes

4. Have the authors made all data underlying the findings in their manuscript fully available?

Reviewer #1: Yes

Reviewer #2: Yes

5. Is the manuscript presented in an intelligible fashion and written in standard English?

Reviewer #1: Yes

Reviewer #2: Yes

6. Review Comments to the Author

Reviewer #1: I really appreciate you solving all the suggestions I provided and work you have done.

1. Check the image quality and format to make sure it satisfy the requirement of the Journal. Use image editing software to check and adjust the resolution.

2. Mention possible interdisciplinary collaborations that could enhance the depth and breadth of subsequent studies.

3. Discuss articles which used similar or relevant solutions to yours. Explicitly highlight the novelty and significance of the new solutions to strengthen your claim. Consider stating more clearly how these solutions differ from those found in previous works and why they are relevant.

Reviewer #2: (No Response)

7. PLOS authors have the option to publish the peer review history of their article (what does this mean?). If published, this will include your full peer review and any attached files.

Reviewer #1: **Yes: **Taolin Qin

Reviewer #2: No

---

## [Author Response · Author response to Decision Letter 1]

1 May 2024

We highly appreciate your detailed and helpful comments on our manuscript, which have helped to significantly improve academic quality and readability of this paper. The revised version has considered all of your recommendations and criticisms.

Reply to reviewer 1: 

1. Check the image quality and format to make sure it satisfies the requirement of the Journal. Use image editing software to check and adjust the resolution.

We have checked the image quality and changed the format under the requirement of the journal.

2. Mention possible interdisciplinary collaborations that could enhance the depth and breadth of subsequent studies.

We have added the subsequent studies in the conclusions, i.e., it is also the limitations of the current manuscript. 

3. Discuss articles which used similar or relevant solutions to yours. Explicitly highlight the novelty and significance of the new solutions to strengthen your claim. Consider stating more clearly how these solutions differ from those found in previous works and why they are relevant.

According to your suggestion, we have revised it as far as we can.

---

## [Editor Report · Decision Letter 2]

6 May 2024

Explicit and exact travelling wave solutions for Hirota equation and Computerized Mechanization

PONE-D-24-02488R2

Dear Dr. Wang,

We’re pleased to inform you that your manuscript has been judged scientifically suitable for publication and will be formally accepted for publication once it meets all outstanding technical requirements.

Kind regards,

Tao Peng

Academic Editor

PLOS ONE
---

## [Editor Report · Acceptance letter]

9 May 2024

PONE-D-24-02488R2 

PLOS ONE

Dear Dr. Wang, 

I'm pleased to inform you that your manuscript has been deemed suitable for publication in PLOS ONE. Congratulations! Your manuscript is now being handed over to our production team.

Kind regards, 

on behalf of

Dr. Tao Peng 

Academic Editor

PLOS ONE